# Mechanical loading modulates phosphate related genes in rat bone

Ashwini Kumar Nepal [1‡], Hubertus W. van Essen[1‡], Christianne M. A. Reijnders[1], Paul Lips[2], Nathalie Bravenboer[1]*

1 Department of Clinical Chemistry, Amsterdam Movement Sciences, Amsterdam UMC, Vrije Universiteit Amsterdam, Amsterdam, The Netherlands, 2 Department of Internal Medicine, Endocrine Section, Amsterdam UMC, Vrije Universiteit Amsterdam, Amsterdam, The Netherlands

‡ AKN and HWE share first authorship on this work.
* n.bravenboer@amsterdamumc.nl

**Data Availability Statement:** All relevant data are within the manuscript and its Supporting Information files.

**Funding:** The author(s) received no specific funding for this work.

## Abstract

Mechanical loading determines bone mass and bone structure, which involves many biochemical signal molecules. Of these molecules, *Mepe* and *Fgf23* are involved in bone mineralization and phosphate homeostasis. Thus, we aimed to explore whether mechanical loading of bone affects factors of phosphate homeostasis. We studied the effect of mechanical loading of bone on the expression of *Fgf23*, *Mepe*, *Dmp1*, *Phex*, *Cyp27b1*, and *Vdr*. Twelve-week old female rats received a 4-point bending load on the right tibia, whereas control rats were not loaded. RT-qPCR was performed on tibia mRNA at 4, 5, 6, 7 or 8 hours after mechanical loading for detection of *Mepe*, *Dmp1*, *Fgf23*, *Phex*, *Cyp27b1*, and *Vdr*. Immunohistochemistry was performed to visualise FGF23 protein in tibiae. Serum FGF23, phosphate and calcium levels were measured in all rats. Four-point bending resulted in a reduction of tibia *Fgf23* gene expression by 64% (p = 0.002) and a reduction of serum FGF23 by 30% (p<0.001), six hours after loading. Eight hours after loading, *Dmp1* and *Mepe* gene expression increased by 151% (p = 0.007) and 100% (p = 0.007). Mechanical loading did not change *Phex*, *Cyp27b1*, and *Vdr* gene expression at any time-point. We conclude that mechanical loading appears to provoke both a paracrine as well as an endocrine response in bone by modulating factors that regulate bone mineralization and phosphate homeostasis.

## Introduction

The ability of the skeleton to form bone after mechanical stimulation is a process that consists of mechanosensing, mechanotransduction and mechanoresponsiveness [1, 2]. Osteocytes function as mechanosensors via canalicular processes and communicating gap junctions in the early stage of bone remodelling [3]. Mechanotransduction involves biochemical signal molecules, which are produced after mechanical loading in a distinct temporal manner. Seconds after mechanical loading, the osteocytic production of factors such as NO and prostaglandins is induced [4]. Within two hours after a single bout of loading, *Cfos* mRNA expression is up

**Competing interests:** The authors have declared
that no competing interests exist.

regulated in the periosteum of the rat tibia [5]. Within a time frame of four hours after loading, osteocyte production of IGF-1 increased [6]. After several months, the cascade of biochemical signalling stimulates osteoblastic bone formation as a mechanoresponsive process that finally results in higher bone mass.

Bone mineralization is one of the various biological processes that govern phosphate homeostasis. Classic regulators of phosphate homeostasis are 1,25-dihydroxyvitamin D3 (1,25 $(OH)_2D_3$) and parathyroid hormone (PTH). In addition, it has been shown that fibroblast growth factor 23 (FGF23) is a regulator of phosphate homeostasis [7–9]. FGF23 is mainly produced by bone cells, such as osteocytes and osteoblasts [10] but recent studies have shown that *Fgf23* expression was also present in bone marrow sinusoidal endothelial cells [11]. FGF23 can be detected in serum in an intact form but also the c-terminal and n-terminal fragments exist as a result of cleavage by furin proteases [12]. FGF23 predominantly exerts its effects in the kidney, where it inhibits the expression of 1-α-hydroxylase (encoded by the gene *Cyp27b1*) and decreases the production of 1,25$(OH)_2D_3$ [13]. Furthermore, FGF23 inhibits the expression of sodium/phosphate co-transporters (NPT2a and NPT2c) that concurrently results in increased phosphate excretion [14]. In the parathyroid gland, FGF23 inhibits PTH production [15, 16]. The production of FGF23 is stimulated by 1,25$(OH)_2D_3$ signalling through the vitamin D receptor (VDR) [13, 17] and by phosphate [18], but in addition appears to be influenced by local factors in bone such as matrix extracellular phosphoglycoprotein (MEPE), Dentin matrix protein-1 (DMP1) and phosphate regulating gene with homologies to endopeptidases on the X-chromosome (PHEX). DMP1, MEPE and PHEX all affect bone mineralization. Initially, DMP1 locally promotes deposition of hydroxyapatite [19, 20]. Cathepsin B is able to cleave a peptide (ASARM peptide) from MEPE, which is known to locally inhibit bone mineralization. Finally, PHEX protein prevents the formation of ASARM peptide [21].

Previous studies revealed that *Mepe* mRNA expression was up-regulated six hours after mechanical loading of the tibia [22] while *Fgf23* gene expression was downregulated six hours after axial compression loading of the ulna [23]. Other investigators have agreed that mechanical loading of bone increased the expression of *Mepe* [24, 25] and *Dmp1* [26]. These findings suggest that bone mineralization can be regulated by mechanical loading, possibly to obtain newly formed bone with optimal quality. However it is not clear whether the immediate effect of mechanical loading on these locally produced factors results in systemic changes in phosphate homeostasis. Therefore we investigated the effect of four-point bending of the rat tibia on gene expression in bone of the phosphate homeostasis related genes *Mepe*, *Dmp1*, *Phex*, *Fgf23*, on the protein expression of FGF23 in bone tissue and on the FGF23 concentration in serum. We also analyzed the gene expression of vitamin D related genes *Vdr* and *Cyp27b1* after mechanical loading. We hypothesised that mechanical loading of bone affects expression of *Fgf23* which results in a change in serum concentration of FGF23.

## Materials and methods

### Experimental animals

Two experiments were performed to detect changes in mRNA and protein expression after mechanical loading. The animal experiments were in accordance with the governmental guidelines for care and use of laboratory animals and approved by the Institutional Animal Care and Use Committee (IACUC) of the VU University Medical Center, Amsterdam, The Netherlands (KE 02–03).

**Experiment A.**   The first experiment aimed to detect the effect of mechanical loading on the mRNA expression of selected genes involved in phosphate homeostasis. Sixty 12-week-old female Wistar rats (Harlan, Horst, The Netherlands), weighing 226 ± 8 grams (mean ± SD)

were randomly assigned to one of six groups (n = 10). The rats in the control group did not receive mechanical loading. The rats in the five load groups received a single bout of mechanical loading and were sacrificed 4, 5, 6, 7 or 8 hours after loading. From all rats, serum was obtained from blood at sacrifice. Moreover, tibiae were dissected and cleaned of soft tissue and periosteum. The proximal and distal ends of the tibia were removed, after which bone marrow was flushed out of the remaining diaphysis with RNAse free water. Lastly, the diaphyses were frozen in liquid nitrogen and stored at -80˚C until RNA extraction.

**Experiment B.**   In the second experiment, we tested whether changes in gene expression would be detectable at the protein level. Thirty-two 12 weeks-old female Wistar rats, weighing 218 ± 15 grams (mean ± SD) were randomly assigned to a single load group of 14 rats, a repeated-load group of 9 rats and a control group of 9 rats. The rats in the load group received a single bout of mechanical loading and were sacrificed 6 hours after loading. The rats in the repeated-load group underwent a single bout of mechanical loading, five days a week, for two weeks. Six hours after the final loading bout, the rats were sacrificed. The rats in the control group received no mechanical loading. The tibiae of all rats in the repeated-load group and five rats in the single load group were dissected, cleaned of soft tissue and prepared for immunohistochemistry. From all rats serum was obtained by cardiac puncture at sacrifice.

## Mechanical loading

One bout of mechanical loading consisted of bending the right tibia of each rat externally in a four-point bending device for 300 cycles with a frequency of 2 Hz and a peak load of 60 N in the single load groups or 50 N in the repeated-load group [6, 27]. Loading was applied under general anaesthesia (2%-2.5% isoflurane/$N_2O/O_2$) and the rats received local subcutaneous analgesia (0.1 mg/kg Temgesic) prior to loading. The left tibia served as a non-loaded contralateral control. At the indicated time points the rats were sacrificed with $CO_2$ gas. A cardiac puncture was performed to collect terminal blood samples, serum was obtained and stored at −20˚C.

## RNA extraction and RT-qPCR

RNA isolation from the tibiae from experiment A was performed by pulverizing the diaphysis with a freezer mill in liquid nitrogen (Spex Certiprep 6750 FreezerMill, Spex Certiprep, Metuchen, NJ, USA). RNA was extracted from bone powder with Trizol for 1 hour at 37˚C and reextracted once with phenol and once with chloroform. Next, the samples were extracted again with Trizol in accordance with the manufacturer's instructions (Invitrogen, Waltham, MA, USA). RNA extracts were incubated with DNAse to eliminate DNA contamination and stored at −80˚C. The absorption at 260 and 280 nm was measured to determine the amount of RNA. A total of 100 ng of total RNA was transcribed into cDNA in a 20 μl mixture containing M-MLV Reverse Transcriptase (Promega, Fitchburg, WI, USA). All RNA samples were assayed in triplicate. Quantitative PCR was performed in a BioRad iCycler Real-time PCR system with three μl of each cDNA sample, 300 nM of each primer and SYBR Green Supermix (BioRad, Hercules, CA, USA) in a total volume of 25 μl. Table 1 lists the primer details of the genes of interest, which were *Fgf23*, *Mepe*, *Dmp1*, *Phex*, *Cyp27b1* and *Vdr* and the housekeeping genes, which were hypoxanthine guanine phosphoribosyl transferase (*Hprt*) and porphobilinogen deaminase (*Pbgd*). After the PCR run a melting curve was conducted from 50˚C to 95˚C to check the specificity of the reactions. Mean Ct-values of each RNA sample assayed in triplicate were used. Gene expression for the genes of interest was normalized for the housekeeping genes ΔCt = (Ct$_{gene of interest}$−Ct$_{housekeeping gene}$) and expressed as fold difference from the average of the housekeeping genes ($2^{-\Delta Ct}$) [28].

**Table 1. Primer sequences for genes used in qPCR analyses.**

| Gene | Accession nr | Primer | Sequence |
|---|---|---|---|
| *Fgf23* | NM_130754 | Forward | 5'-GAT-GCT-GGC-TCC-GTA-GTG-AT-3' |
| | | Reverse | 5'-CGT-CGT-AGC-CGT-TCT-CTA-GC-3' |
| *Dmp1* | NM_203493 | Forward | 5'-GCG-ACT-CCA-CAG-AGG-ATT-TC-3' |
| | | Reverse | 5'-GTC-CCT-CTG-GGC-TAT-CTT-CC-3' |
| *Mepe* | NM_024142 | Forward | 5'-AAG-ACA-AGC-CAC-CCT-ACA-CG-3' |
| | | Reverse | 5'-CCC-ACT-GGA-TGA-TGA-CTC-ACT-3' |
| *Phex* | NM_013004 | Forward | 5'-CAG-GCA-TCA-CAT-TCA-CCA-AC-3' |
| | | Reverse | 5'-GGA-GGA-CTG-TGA-GCA-CCA-AT-3' |
| *Cyp27b1* | NM_053763 | Forward | 5'-CCC-GAC-ACA-GAA-ACC-TTC-AT-3' |
| | | Reverse | 5'-GGC-AAA-CAT-CTG-ATC-CCA-GT-3' |
| *Vdr* | NM_017058 | Forward | 5'-ACA-GTC-TGA-GGC-CCA-AGC-TA-3' |
| | | Reverse | 5'-TCC-CTG-AAG-TCA-GCG-TAG-GT-3' |
| *Hprt* | X62085 | Forward | 5'- GTG-TCA-TCA-GCG-AAA-GTG-GA-3' |
| | | Reverse | 5'- TAC-TGG-CCA-CAT-CAA-CAG-GA-3' |
| *Pbgd* | Y12006 | Forward | 5'- ATG-TCC-GGT-AAC-GGC-GGC-3' |
| | | Reverse | 5'- CAA-GGT-TTT-CAG-CAT-CGC-TAC-CA- 3' |

## Tissue preparation and immunohistochemistry for FGF23

After dissection, the condyles at the proximal end of the tibiae were sawn off with a diamond saw and the distal end was cut off with scissors. The tibiae were immediately fixed for 24 hours at 4˚C in 4% paraformaldehyde in phosphate buffered saline (PBS). After fixation the tibiae were decalcified in 10% EDTA and 0.5% paraformaldehyde in PBS at 4˚C for 4 weeks. Proper decalcification was verified by x-ray photography. The tibiae were then dehydrated and embedded in paraffin and cut into 5 μm thick sections. Sections were rehydrated and endogenous peroxidase was quenched with 3% $H_2O_2$ in 40% methanol/PBS. Antigen retrieval was performed through incubation with 0.5% trypsin + 0.1% CaCl2 for 15 minutes at 37˚C. After the blocking of the non-specific binding sites with blocking solution (Invitrogen, Waltham, MA, USA) for 1 hour, the sections were incubated overnight at 4˚C with 1/1000 anti-intact FGF23 antibody (Santa Cruz Biotechnology Inc, Dallas, TX, USA). The sections were incubated with 1/100 biotinylated second antibody (Dako, Agilent Technologies, Inc., Santa Clara, CA, USA). Further enhancement was performed with the Tyramide Signal Amplification kit (Invitrogen, Waltham, MA, USA) and color development was performed with AEC (Zymed technologies, Waltham, MA, USA). The sections were counterstained with Mayer's hematoxylin. Quantitative evaluation of the specific staining for FGF23 was performed with NIS Elements digital imaging software (Nikon, Tokyo, Japan). Digital images were made from the medial cortex of two sections of each tibia at 10x magnification. The area of positively-stained osteocytes, the area of positively-stained capillairy bloodvessels and the total bone area was measured. The percentage of positively-stained bone area and the percentage of positively stained vessel area were calculated. Data from the two sections of each tibia were averaged.

## Serum analysis

In sera of all rats from the two experiments, intact FGF23 concentration was measured by ELISA (Kainos Laboratories, Tokyo, Japan) as described in Heijboer et al [29]. Serum inorganic phosphorus and calcium were analyzed using a Modular P800 automatic analyzer (Roche, Mannheim, Germany).

## Statistical analysis

For statistical analysis of qPCR expression, data were log transformed to obtain a normal distribution. Differences in expression between the loaded tibiae and the non-loaded tibiae in all groups were tested with one-way analysis of variance (ANOVA) and Dunnet's post-hoc test was used to test for significant differences between the time groups and the control group. A non-parametric paired t-test (Wilcoxon) was used to test for significant differences in immunohistochemical staining. Statistical analysis of the serum data was performed with one-way ANOVA. Results were considered significantly different at $p < 0.05$. Graphpad 4.0 (Graphpad Software, San Diego, CA, USA) was used for statistical analysis.

## Results

### Gene expression

Experiment A evaluated the effect of mechanical loading on gene expression (Fig 1A–1F) shows the normalized mRNA expression of, *Mepe*, *Dmp1*, *Fgf23*, *Phex*, *Cyp27b1* and *Vdr* in the loaded and non-loaded tibiae. To avoid the possibility that systemic loading effects on the left non-loaded tibia would interfere with the analysis, the normalized expression of the right-loaded tibiae in the time groups was compared with the normalized expression in the right-control tibiae. The expression of *Mepe* and *Dmp1* was increased at 5, 6 and 8 hours after mechanical loading, which was at a maximum of 103% for *Mepe* (p = 0.007) and 151% for *Dmp1* (p = 0.007) eight hours after loading (Fig 1A and 1B). *Fgf23* gene expression was decreased at 4, 6 and 7 hours after mechanical loading and the maximum reduction was 64%

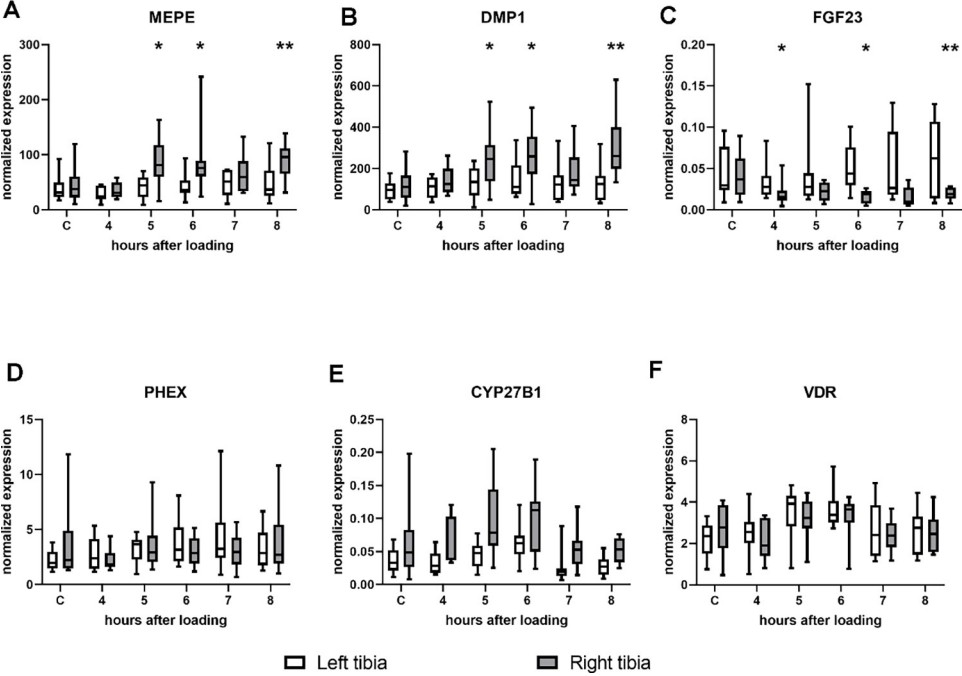

**Fig 1. (A-F):** Box whisker plots of gene expression of *Mepe*, *Dmp1*, *Fgf23*, *Phex*, *Cyp27b1* and *Vdr* in the contra-lateral control (left) and loaded tibiae (right). Rats underwent a single bout of mechanical loading of the right tibia and were sacrificed after the indicated time. Rats in the control group underwent no mechanical loading. The box represents median, 25th and 75th percentiles, and the whiskers represent highest and lowest gene expression values. For statistical analysis the data were log transformed. *p<0.05 versus right tibia in the control group. **p<0.01 versus right tibia in the control group.

(p = 0.002), 7 hours after loading (Fig 1C). The expression of the other genes, *Phex*, *Vdr* and *Cyp27b1* was not different in the right-loaded tibiae of the time groups compared to the expression in the right control tibia (Fig 1D–1F). No difference was observed in gene expression for any of the tested genes when the left tibiae from the loaded and control groups were compared.

## Immunohistochemistry

To test whether the effect on gene expression was also detectable at the protein level, FGF23 protein was quantified in bone tissue by immunohistochemistry. FGF23 positive staining was observed in the osteocytes of the cortex and in the capillary blood vessels within the cortical bone. Weak staining of osteoblasts on the endocortical side of the cortex was detected. Osteocytes in trabecular bone only sporadically showed positive staining. Measurement of the area of FGF23 positive staining in osteocytes in the cortical bone of the tibiae did not reveal significant differences between the loaded tibiae and the contra-lateral control tibiae after single loading or after repeated loading. Likewise there was no significant difference in FGF23 positive staining of the capillary blood vessels between the loaded tibiae and the contralateral control tibiae. (Fig 2A–2E).

## Serum analysis

To further investigate the effect of mechanical loading on protein expression of FGF23, serum concentrations of FGF23, phosphate and calcium were measured in the control rats, the single-loaded rats and the repeated-loaded rats. Six hours after a single bout of mechanical loading, serum FGF23 concentration reduced significantly in loaded rats, compared to control rats (110 ± 7 pg/mL vs 157 ± 9 pg/mL: p<0.001) (Fig 3A). Serum FGF23 concentration did not change in repeated-loaded rats, compared to control rats (153 ± 10 pg/mL). Mechanical loading did not change the serum concentrations of inorganic phosphate (Fig 3B) and calcium (data not shown).

## Discussion

Our results demonstrated that mechanical loading increased expression of *Mepe* and *Dmp1*, and decreased *Fgf23*, whereas mechanical loading showed no effects on expression of *Phex*, *Vdr* and *Cyp27b1*. FGF23 protein expression in bone and in capillary blood vessels measured by immunohistochemistry was not affected six hours after loading nor after repeated loading for 10 days. Loading decreased FGF23 in serum at 6 hours after loading, however, this decrease did not persist after repeated loading for 10 days. Loading did not affect serum levels of phosphate and calcium.

The decreased expression of *Fgf23* confirms earlier results in a rat axial ulna loading study where a decrease of *Fgf23* expression 6 hours after a single bout of loading was found [23]. There are only a few studies that investigated the effect of mechanical loading on the expression of *Fgf23* and phosphate homeostasis. These studies have performed controlled mechanical loading in the form of axial compression of the rat ulna [30] or axial compression of the mouse tibia [31]. They reported that gene expression of *Fgf23* decreased after several days of repeated mechanical loading. A different mouse training model was performed by Gardinier et al [32]. Here the mice exercised 30 minutes every day in a treadmill. *Fgf23* gene expression was increased after 6 days of training, but not after 2 or 4 days of training.

We observed a strong staining for FGF23 in osteocytes in the cortex of the tibia by immunohistochemistry. However, strong staining was also observed in capillary blood vessels within the cortical bone. This could represent FGF23 being secreted by osteocytes into the circulation,

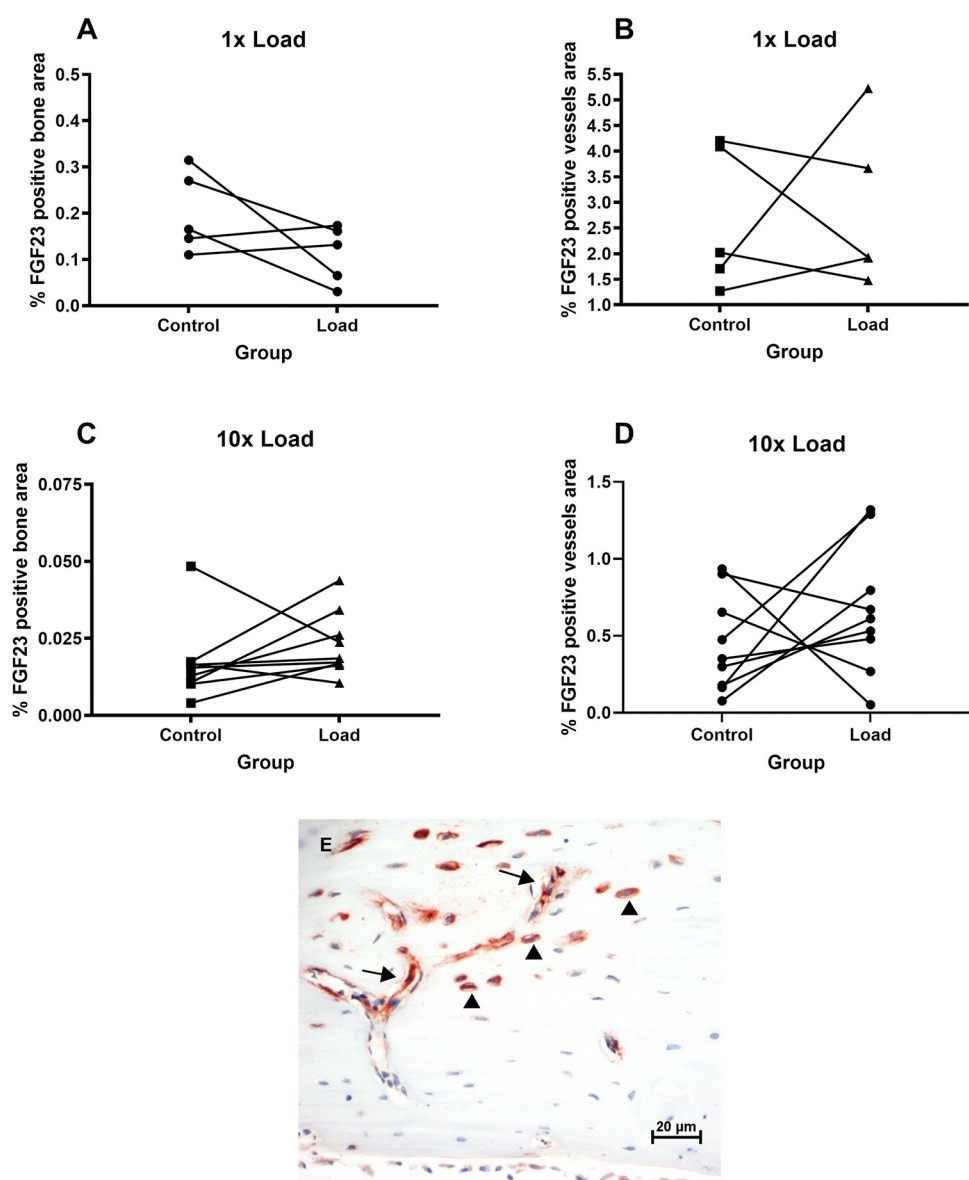

**Fig 2. (A-E):** Quantitative analysis of immunohistochemistry. Rats received a single bout of mechanical loading of the right tibia (A and B) or repeated loading of the right tibia (C and D). Paraffin sections of the tibiae were cut and stained for FGF23. The FGF23 positive area in as a percentage of the total bone area was measured in osteocytes (A and C) and in capillary blood vessels (B and D). (E) Representative image of FGF23 positive staining in osteocytes (arrowheads) and in capillary blood vessels (arrows).

but it could also confirm reports showing FGF23 expression in endothelial cells of small blood vessels [11]. We did not find an effect of mechanical loading on the protein expression in cortical bone, both in osteocytes and in capillary blood vessels. The discrepancy between mRNA and protein results could be due to methodological limitations. Immunohistochemistry is at best a semi-quantitative method. Automatically-quantified positive area was used to measure bone FGF23 protein expression. Other methods, such as counting positive cells or measuring the color density, might be more accurate but are technically still unfeasible. Moreover, immunohistochemical staining for FGF23 exhibited large variation, which suggests that a larger number of animals might be needed to detect local bone changes in FGF23 protein.

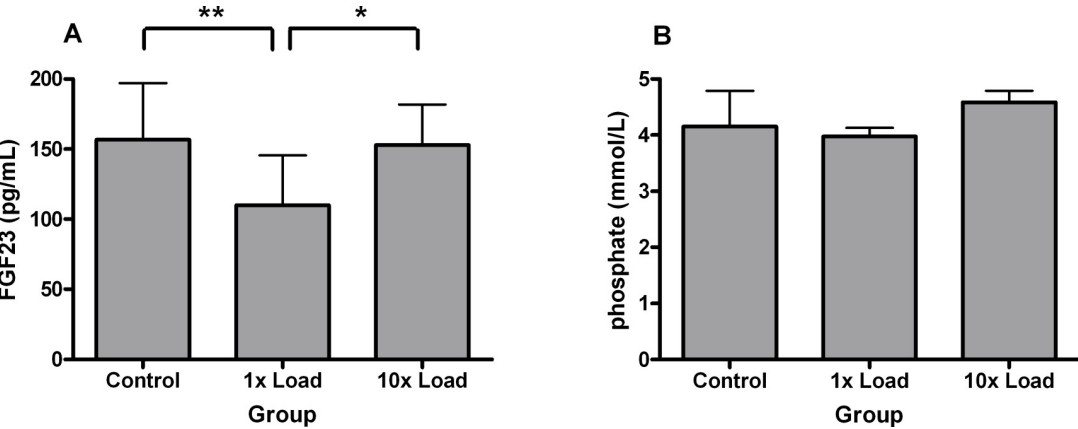

**Fig 3. (A and B):** Serum concentration of intact FGF23 (A) and inorganic phosphate (B). Rats in the control group (n = 19) received no mechanical loading. Rats in the 1x Load group (n = 24) received a single bout of mechanical loading of the right tibia and were sacrificed 6 hours later. Rats in the 10x Load group (n = 9) received a single bout of mechanical loading of the right tibia every workday for two weeks and were sacrificed 6 hours after the last bout of mechanical loading. Blood samples were collected at sacrifice and FGF23 and inorganic phosphate were measured in serum. Values are means ± SD. \*\*p<0.001.

A significant decrease of the FGF23 concentration in serum was observed six hours after mechanical loading but this decrease in serum FGF23 was not observed after 10 days of repeated loading. Our results suggest that the changes in mechanical loading of the skeleton quickly and substantially affected systemic FGF23. A comparably quick response in serum FGF23 has been shown after injection of mice with 1,25(OH)2D. This resulted in an 1.6-fold increase in serum FGF23 concentration within four hours [33]. Likewise, a single injection of mice with erythropoietin can also lead to a 2–4 times increase in serum intact FGF23 within 4–6 hours after injection [34]. Nevertheless, the systemic changes observed in serum FGF23 by loading of a single tibia only at 6 hours could be arbitrary and not an outcome of loading. Contrary to our expectations, it was observed that when bone was subjected to two weeks of daily mechanical loading, the serum FGF23 concentration was not different from controls, suggesting activation of a negative feedback mechanism to restore the serum FGF23 concentration to its original level after its initial decrease [14]. Lower levels of serum FGF23 would increase phosphate retention [9]. Nevertheless we did not find a change in serum phosphate concentration. A likely explanation would be that negative feedback mechanisms could be activated to maintain serum phosphate concentration within narrow boundaries by modulating bone metabolism, phosphate absorption in the gut and phosphate excretion by the kidney [35]. Hence, changes in phosphate metabolism through actions of FGF23 are possibly detectable more precisely in urinary phosphate excretion than in serum phosphate levels.

In the present study, mechanical loading increased the expression of *Mepe* and *Dmp1* while the expression of *Phex* did not change. This confirms earlier findings that *Mepe* [22, 24, 25] and *Dmp1* [20, 26] expression was increased after mechanical loading. Increased DMP1 production promotes bone mineralization. However, increased MEPE/PHEX ratio might lead to a higher formation of ASARM peptides and these are associated with inhibition of the mineralization process. DMP1 and PHEX are known to inhibit FGF23 production [36]. A single bout of mechanical loading decreased *Fgf23* mRNA expression as early as 4 hours after mechanical loading while increases of *Mepe* and *Dmp1* started at 5 hours after loading. This suggests that the change in *Fgf23* expression was not the result of increases in MEPE or DMP1 expression.

One finding is that mechanical loading did not change the expression of *Cyp27b1* and *Vdr* within the studied time frame of 8 hours. This is in line with previous results in which we did

not find a differences in *Cyp27b1* expression, 6 hours after in vivo axial mechanical loading of the ulna [23]. However, an in vitro study reported a 2-fold increase in *Cyp27b1* and 0.6 fold decrease in *Vdr*, 3 hours after mechanical loading, applied as a pulsating fluid flow [37]. 1,25 (OH)2D interacts with mechanical loading in vitro at the level of mechanotransduction, in which modulation of the mechanoresponse occurs via a mechanism which is independent of VDR genomic actions [38].

We demonstrated that a single bout of mechanical loading decreased gene expression of *Fgf23* in the loaded bone and at the same time decreased serum levels of FGF23. However we could not detect changes in protein expression by immunohistochemistry in the loaded bone directly after loading. The most obvious reason is that changes in protein production in the osteocytes and endothelial cells will be detectable at a later time point, but it is also possible that downstream processing of mRNA is influenced by mechanical loading. We did see a decrease in FGF23 concentration is serum. The ELISA method we performed to measure FGF23 in serum detects only intact FGF23. Therefore the decrease that was detected six hours after loading could be caused by a genuine quick decrease in protein production by the osteocytes but also by increased cleavage of the intact protein. The responses to acute mechanical loading seem to have disappeared when loading was applied for 10 days in a row. This may be the result of a feedback mechanism, possibly involving 1,25(OH)2D [17].

A limitation of this study is that we only performed immunohistochemistry in the repeated-loading experiment and we did not have a group for RNA extraction from bone, therefore we could not test *Fgf23* gene expression after ten bouts of mechanical loading. A second limitation of this study is that we did not measure phosphate concentration in urine. With the changes in FGF23 concentration one would expect changes in phosphate retention but we could not conclude this from the serum phosphate results. Urinary phosphate concentration would more likely reflect changes in phosphate homeostasis. In this study we showed changes in phosphate related gene expression for *Mepe*, *Fgf23* and *Dmp1* at different time points after loading and decreased serum FGF23, 6 hours after loading, these results do not show a direct effect on bone mineralization, as bone adaptation is expected after days or weeks and not within the timeframe of this study.

Taken together, this study shows that mechanical loading of bone quickly decreases *Fgf23* gene expression and simultaneously decreases serum FGF23 concentration. Decreased FGF23 can lead to retention of phosphate and increased production of 1,25(OH)2D. This indicates that mechanical loading plays a role in the maintenance of serum phosphate concentration. We demonstrated that mechanical loading induces alterations in expression of *Mepe* and *Dmp1*, which suggests that there are local influences of mechanical loading on bone mineralization. In conclusion, mechanical loading appears to provoke both paracrine as well as endocrine response in bone by modulating factors which regulate bone mineralization and phosphate homeostasis.

## Supporting information

**S1 File. Result Immunhistochemistry.** Percentage of FGF23 positive area in bone and vessels after single load or repeated load.
(XLSX)

**S2 File. Result gene expression.** Normalized expression of measured genes in the left and right tibias at the timepoints.
(XLSX)

**S3 File. Result serum measurements.** Serum concentrations of FGF23, phosphate and calcium.
(XLSX)

## Acknowledgments

The authors thank Josien Dijkstra-Lagemaat for excellent technical assistance with the determination of FGF23 concentration in serum.

## Author Contributions

**Conceptualization:** Ashwini Kumar Nepal, Hubertus W. van Essen, Christianne M. A. Reijnders, Paul Lips.

**Formal analysis:** Ashwini Kumar Nepal, Hubertus W. van Essen, Christianne M. A. Reijnders, Paul Lips, Nathalie Bravenboer.

**Funding acquisition:** Paul Lips.

**Methodology:** Ashwini Kumar Nepal, Hubertus W. van Essen, Christianne M. A. Reijnders, Paul Lips, Nathalie Bravenboer.

**Resources:** Nathalie Bravenboer.

**Supervision:** Paul Lips, Nathalie Bravenboer.

**Writing – original draft:** Ashwini Kumar Nepal, Hubertus W. van Essen, Christianne M. A. Reijnders, Paul Lips.

**Writing – review & editing:** Ashwini Kumar Nepal, Hubertus W. van Essen, Christianne M. A. Reijnders, Paul Lips, Nathalie Bravenboer.

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
