## [Decision Letter · Decision Letter 0]

26 Sep 2022

PONE-D-22-21743Mechanical loading modulates phosphate related genes in rat bonePLOS ONE

Dear Dr. Bravenboer,

Thank you for submitting your manuscript to PLOS ONE. After careful consideration, we feel that it has merit but does not fully meet PLOS ONE’s publication criteria as it currently stands. Therefore, we invite you to submit a revised version of the manuscript that addresses the points raised during the review process.

We look forward to receiving your revised manuscript.

Kind regards,

Damian Christopher Genetos, Ph.D.

Academic Editor

PLOS ONE

Journal Requirements:

Reviewers' comments:

Reviewer's Responses to Questions

**Comments to the Author**

1. Is the manuscript technically sound, and do the data support the conclusions?

Reviewer #1: Yes

2. Has the statistical analysis been performed appropriately and rigorously? 

Reviewer #1: Yes

3. Have the authors made all data underlying the findings in their manuscript fully available?

Reviewer #1: Yes

4. Is the manuscript presented in an intelligible fashion and written in standard English?

Reviewer #1: Yes

5. Review Comments to the Author

Reviewer #1: Comments to authors:

In this manuscript, Nepal et al investigate the link between mechanical loading of bone and alterations in FGF23, an osteocyte gene linked to the regulation of phosphate homeostasis. The authors employed a 4-point tibial bending model in female rats and examined phosphate- and vitamin d-related gene expression along a short-term time course, and also measured FGF23 in the serum and by immunostaining. Key results show that FGF23 gene expression in de-marrowed bone and protein levels in the serum were down-regulated by loading, although neither phosphate nor calcium levels in the circulation were affected.

In general, the studies are quite simple in nature but do shed new light on the potential link between mechanical loading and systemic phosphate homeostasis, presumably needed to mineralize newly formed osteoid. The manuscript would be strengthened if the authors could support their findings by referencing one of the many RNA-Seq or microarray-based studies now widely available, seeing if other laboratories have linked any of these gene changes to altered mechanical loading. Moreover, a better explanation for why FGF23 protein levels did not change in the bone, nor phosphate levels in the serum, despite altered FGF23 mRNA levels is warranted. These and other comments are expressed in detail below.

1. Introduction, page 4, line 69: The authors state here that FGF23 is primarily produced by bone cells such as osteocytes, but newly emerging research suggests that appreciable FGF23 expression can be achieved by marrow sinusoid endothelial cells, adipocytes, and erythroid cells (see abstracts # 1007, 1008, and 1110 from the 2022 American Society for Bone and Mineral Research annual meeting proceedings). Therefore, the authors are encouraged to consider these new data, and place their findings from primarily osteocyte-enriched tissue in this newly emerging context. This is also relevant to the Discussion section, page 15, lines 305-307, as the authors observed strong FGF23 staining in capillary blood vessels in their samples.

2. Introduction, page 5, lines 91 and onwards: As the authors already nicely summarize what’s known about MEPE and other phosphate regulatory genes, a stronger effort needs to be made here emphasizing what is novel about this paper. For example, if FGF23 has never been shown to be mechanically regulated, this would be a novel question to address.

3. Results, page 11, lines 201-202: While it is commendable that the authors have used non-loaded mice as controls for their bending loaded mice, as phosphate regulation is systemic, I think it would be of strong interest to the reader to also look at comparisons between the loaded and non-loaded limbs. This is important for other scientists who want to build on this work: Is there a systemic effect visible in the non-loaded limb, and if so, what does that mean for all of these previous studies that use a non-loaded contralateral? If these datasets from the non-loaded limbs are available, the authors are strongly encouraged to add them (at least as a supplement) and interpret them accordingly.

4. Results, page 12, lines 219-227: As the authors have used a tibial bending model, and noted that FGF23 expression via immunostaining was strongest in the cells nearest the cortext, this raises the question: Did the authors observe any changes in expression related to the neutral axis for bending? If they confined their quantitative analyses to the points furthest from the neutral axis, would a loading-induced trend emerge? Please address.

5. Results, general: As the results of this study are fairly simple in nature, the authors are encouraged to look deeper into the literature, paying close attention to previous studies that conducted mechanical loading analysis of osteocytes in vitro or of bone in vivo and conducted RNA-Seq or other unbiased analyses (as one such example: PMID: 26573903, although I’m sure many others exist). Do FGF23 or any other phosphate regulatory genes show up in these datasets? Knowing if this occurs could strengthen the arguments in the current manuscript, which is lacking much in the way of mechanism.

6. Results, general: A major limitation of this study is that urine samples were not collected to measure phosphate levels being excreted; this needs to be acknowledged somewhere in the Discussion section.

7. Discussion, page 13, lines 258-260: This sentence needs further explanation, as it is difficult to follow the authors’ logic at present: “Loading decreased FGF23 in serum at 6 hours, however this decrease did not persist after 10 days of continuous loading, which suggests that mechanical loading may provoke an endocrine effect”. What endocrine effect, and what systems are involved? How would this explain why FGF23 changed in acute but not chronic loading?

6. PLOS authors have the option to publish the peer review history of their article (what does this mean?). If published, this will include your full peer review and any attached files.

Reviewer #1: No

---

## [Author Response · Author response to Decision Letter 0]

16 Jan 2023

Response to reviewers

Reviewer #1: Comments to authors:

In this manuscript, Nepal et al investigate the link between mechanical loading of bone and alterations in FGF23, an osteocyte gene linked to the regulation of phosphate homeostasis. The authors employed a 4-point tibial bending model in female rats and examined phosphate- and vitamin d-related gene expression along a short-term time course, and also measured FGF23 in the serum and by immunostaining. Key results show that FGF23 gene expression in de-marrowed bone and protein levels in the serum were down-regulated by loading, although neither phosphate nor calcium levels in the circulation were affected.

In general, the studies are quite simple in nature but do shed new light on the potential link between mechanical loading and systemic phosphate homeostasis, presumably needed to mineralize newly formed osteoid. The manuscript would be strengthened if the authors could support their findings by referencing one of the many RNA-Seq or microarray-based studies now widely available, seeing if other laboratories have linked any of these gene changes to altered mechanical loading. Moreover, a better explanation for why FGF23 protein levels did not change in the bone, nor phosphate levels in the serum, despite altered FGF23 mRNA levels is warranted. These and other comments are expressed in detail below.

Answer: The authors would like to thank the editorial team and the reviewer for their thoughtful suggestions on the manuscript. We have provided response to each comments below, and the changes are also reflected in the manuscript text. 

1. Introduction, page 4, line 69: The authors state here that FGF23 is primarily produced by bone cells such as osteocytes, but newly emerging research suggests that appreciable FGF23 expression can be achieved by marrow sinusoid endothelial cells, adipocytes, and erythroid cells (see abstracts # 1007, 1008, and 1110 from the 2022 American Society for Bone and Mineral Research annual meeting proceedings). Therefore, the authors are encouraged to consider these new data, and place their findings from primarily osteocyte-enriched tissue in this newly emerging context. This is also relevant to the Discussion section, page 15, lines 305-307, as the authors observed strong FGF23 staining in capillary blood vessels in their samples.

Answer: These new findings are very interesting. We do find a reasonable amount of staining for FGF23 in the capillary blood vessels which could be locally produced by endothelial cells. As the gene expression was measured in RNA extracted from the total cortical tibia, including the capillary blood vessels the changes after mechanical loading we cannot distinguish between osteocytes and endothelial cells. We do not know whether the gene expression by sinusoid capillary blood vessels is responsive to mechanical loading in the same way as osteocytes. We do not find an effect of mechanical loading on the protein expression in osteocytes and in sinusoidal capillary endothelial cells. In this version of the manuscript we have included the full data of the immunohistochemistry. 

We added a reference to this new information in the introduction (line 77 “FGF23 is mainly produced by bone cells, such as osteocytes and osteoblasts [10] but recent studies have shown that Fgf23 expression was also present in bone marrow sinusoidal endothelial cells [11].”) and in the discussion (line 285 – 288 “However, strong staining was also observed in capillary blood vessels within the cortical bone. This could represent FGF23 being secreted by osteocytes into the circulation, but it could also confirm reports showing FGF23 expression in endothelial cells of small blood vessels [11].”).

2. Introduction, page 5, lines 91 and onwards: As the authors already nicely summarize what’s known about MEPE and other phosphate regulatory genes, a stronger effort needs to be made here emphasizing what is novel about this paper. For example, if FGF23 has never been shown to be mechanically regulated, this would be a novel question to address.

Answer: The field of research on phosphate homeostasis and bone is rapidly evolving. We have changed the focus of the manuscript more towards FGF23 and the protein expression of FGF23. We think that the fact that changes in gene expression are immediately visible in changes in serum FGF23 is the novel message in the manuscript.

In the introduction we added a stronger emphasis on the need to analyze protein expression of FGF23 (line 99 – 102 “However it is not clear whether the immediate effect of mechanical loading on these locally produced factors results in systemic changes in phosphate homeostasis. Therefore we investigated the effect of four-point bending of the rat tibia on gene expression in bone of the phosphate homeostasis related genes Mepe, Dmp1, Phex, Fgf23, on the protein expression of FGF23 in bone tissue and on the FGF23 concentration in serum.”). The discussion has been re-written to discuss the total change in expression of FGF23 from mRNA to serum concentration (line 334 – 345).

3. Results, page 11, lines 201-202: While it is commendable that the authors have used non-loaded mice as controls for their bending loaded mice, as phosphate regulation is systemic, I think it would be of strong interest to the reader to also look at comparisons between the loaded and non-loaded limbs. This is important for other scientists who want to build on this work: Is there a systemic effect visible in the non-loaded limb, and if so, what does that mean for all of these previous studies that use a non-loaded contralateral? If these datasets from the non-loaded limbs are available, the authors are strongly encouraged to add them (at least as a supplement) and interpret them accordingly.

Answer: In this experiment and in previous pilot experiments in our lab using 4-point bending as loading device, we have seen that the variation in gene expression of mechanosensitive genes between the right and left leg of non-loaded control rats is similar to the variation between the same legs of different non-loaded control rats. The aim of our study was to find systemic effects of mechanical loading. When systemic effects of a local intervention are expected or investigated the use of non-loaded contralateral controls are not advised. For this reason we have decided to only test the right legs. We will add the full dataset of the gene expression results as supplement. 

4. Results, page 12, lines 219-227: As the authors have used a tibial bending model, and noted that FGF23 expression via immunostaining was strongest in the cells nearest the cortex, this raises the question: Did the authors observe any changes in expression related to the neutral axis for bending? If they confined their quantitative analyses to the points furthest from the neutral axis, would a loading-induced trend emerge? Please address.

Answer: The measurements were performed in longitudinal sections of the medial part of the tibia. We counted positive osteocytes in one cortex from the proximal end to the distal end of the long bone. We used standardized procedures for embedding, sectioning and mounting to ensure we measured the same cortical side of the tibia from each animal. Four point bending will lead to compression on the upper side of the bone and tension on the lower side of the bone. We assumed the neutral axis would fall within the marrow space. Then all the osteocytes in the upper cortex will perceive compression stress and all the osteocytes in the lower cortex will perceive tension stress.

5. Results, general: As the results of this study are fairly simple in nature, the authors are encouraged to look deeper into the literature, paying close attention to previous studies that conducted mechanical loading analysis of osteocytes in vitro or of bone in vivo and conducted RNA-Seq or other unbiased analyses (as one such example: PMID: 26573903, although I’m sure many others exist). Do FGF23 or any other phosphate regulatory genes show up in these datasets? Knowing if this occurs could strengthen the arguments in the current manuscript, which is lacking much in the way of mechanism.

Indeed there have been a number of studies on mechanical loading of bone tissue in vivo and mechanical loading of bone cells in vitro which employ RNA-seq or microarrays in which the response of phosphate regulatory are mentioned. In several in vivo studies FGF23 gene expression was decreased, albeit after 3-12 days of repeated loading. These studies did not focus on phosphate metabolism and did not provide an explanation for the decreased expression of FGF23. We have referred to these studies in the text. In vitro studies used the MLO-Y4 cell line which does not express FGF23.

In the discussion we refer to some of the more recent studies (line 277 – 283 “There are only a few studies that investigated the effect of mechanical loading on the expression of Fgf23 and phosphate homeostasis. These studies have performed controlled mechanical loading in the form of axial compression of the rat ulna [30] or axial compression of the mouse tibia [31]. They reported that gene expression of Fgf23 decreased after several days of repeated mechanical loading. A different mouse training model was performed by Gardinier et al [32]. Here the mice exercised 30 minutes every day in a treadmill. Fgf23 gene expression was increased after 6 days of training, but not after 2 or 4 days of training.”).

6. Results, general: A major limitation of this study is that urine samples were not collected to measure phosphate levels being excreted; this needs to be acknowledged somewhere in the Discussion section.

Answer: We only received ethical clearance to group-house the rats. The urine collection of rats requires 16-24 hours of single housing of rats in metabolic cages, which was not possible in this study due to ethical reasons. 

In the discussion we have named this as one of the limitations of this study (line 347 – 350 “A second limitation of this study is that we did not measure phosphate concentration in urine. With the changes in FGF23 concentration one would expect changes in phosphate retention but we could not conclude this from the serum phosphate results. Urinary phosphate concentration would more likely reflect changes in phosphate homeostasis.”).

7. Discussion, page 13, lines 258-260: This sentence needs further explanation, as it is difficult to follow the authors’ logic at present: “Loading decreased FGF23 in serum at 6 hours, however this decrease did not persist after 10 days of continuous loading, which suggests that mechanical loading may provoke an endocrine effect”. What endocrine effect, and what systems are involved? How would this explain why FGF23 changed in acute but not chronic loading?

Answer: We agree with the reviewer this sentence is unclear. We have removed this sentence from the text.

---

## [Decision Letter · Decision Letter 1]

21 Feb 2023

Mechanical loading modulates phosphate related genes in rat bone

PONE-D-22-21743R1

Dear Dr. Bravenboer,

We’re pleased to inform you that your manuscript has been judged scientifically suitable for publication and will be formally accepted for publication once it meets all outstanding technical requirements.

Kind regards,

Damian Christopher Genetos, Ph.D.

Academic Editor

PLOS ONE

Additional Editor Comments (optional):

Reviewers' comments:

Reviewer's Responses to Questions

**Comments to the Author**

1. If the authors have adequately addressed your comments raised in a previous round of review and you feel that this manuscript is now acceptable for publication, you may indicate that here to bypass the “Comments to the Author” section, enter your conflict of interest statement in the “Confidential to Editor” section, and submit your "Accept" recommendation.

Reviewer #1: All comments have been addressed

2. Is the manuscript technically sound, and do the data support the conclusions?

Reviewer #1: Yes

3. Has the statistical analysis been performed appropriately and rigorously? 

Reviewer #1: Yes

4. Have the authors made all data underlying the findings in their manuscript fully available?

Reviewer #1: Yes

5. Is the manuscript presented in an intelligible fashion and written in standard English?

Reviewer #1: Yes

6. Review Comments to the Author

Reviewer #1: (No Response)

7. PLOS authors have the option to publish the peer review history of their article (what does this mean?). If published, this will include your full peer review and any attached files.

Reviewer #1: No

---

## [Editor Report · Acceptance letter]

27 Feb 2023

PONE-D-22-21743R1 

Mechanical loading modulates phosphate related genes in rat bone 

Dear Dr. Bravenboer:

I'm pleased to inform you that your manuscript has been deemed suitable for publication in PLOS ONE. Congratulations! Your manuscript is now with our production department. 

Kind regards, 

on behalf of

Dr. Damian Christopher Genetos 

Academic Editor

PLOS ONE